# Impact of the pandemic and concomitant COVID-19 on the management and outcomes of middle cerebral artery strokes: a nationwide registry-based study

Abdul Karim Ghaith,[1] Victor Gabriel El-Hajj [ID],[2] Elias Atallah,[3] Jorge Rios Zermeno [ID],[4] Krishnan Ravindran,[4] Maria Gharios,[2] Harry Hoang,[1] Mohamad Bydon,[1] Marcus Ohlsson,[2] Adrian Elmi-Terander [ID],[2] Rabih G Tawk,[4] Pascal Jabbour[5]

AKG and VGE-H contributed equally.

For numbered affiliations see end of article.

**Correspondence to**
Dr Adrian Elmi-Terander;
adrian.elmi.terander@ki.se

## ABSTRACT

**Objectives** To investigate the impact of the COVID-19 pandemic as well as concomitant COVID-19 itself on stroke care, focusing on middle cerebral artery (MCA) territory infarctions.

**Design** Registry-based study.

**Setting** We used the National Inpatient Sample (NIS) database, which covers a wide range of hospitals within the USA.

**Participants** The NIS was queried for patients with MCA strokes between 2016 and 2020. In total, 35 231 patients were included.

**Outcome measures** Outcome measures were postprocedural complications, length of stays (LOSs), in-hospital mortality and non-routine discharge. Propensity score matching using all available baseline variables was performed to reduce confounders when comparing patients with and without concomitant COVID-19.

**Results** Mechanical thrombectomy (MT) was performed in 48.4%, intravenous thrombolysis (IVT) in 38.2%, and both MT and IVT (MT+IVT) in 13.4% of patients. A gradual increase in the use of MT and an opposite decrease in the use of IVT (p<0.001) was detected during the study period. Overall, 25.0% of all patients were admitted for MCA strokes during the pandemic period (2020), of these 209 (2.4%) were concomitantly diagnosed with COVID-19. Patients with MCA strokes and concomitant COVID-19 were significantly younger (64.9 vs 70.0; p<0.001), had significantly worse NIH Stroke Severity scores, and worse outcomes in terms of LOS (12.3 vs 8.2; p<0.001), in-hospital mortality (26.3% vs 9.8%; p<0.001) and non-routine discharge (84.2% vs 76.9%; p=0.013), as compared with those without COVID-19. After matching, only in-hospital mortality rates remained significantly higher in patients with COVID-19 (26.7% vs 8.5%; p<0.001). Additionally, patients with COVID-19 had higher rates of thromboembolic (12.3% vs 7.6%; p=0.035) and respiratory (11.3% vs 6.6%; p=0.029) complications.

**Conclusions** Among patients with MCA stroke, those with concomitant COVID-19 were significantly younger and had higher stroke severity scores. They were more likely to experience thromboembolic and respiratory complications and in-hospital mortality compared with matched controls.

### STRENGTHS AND LIMITATIONS OF THIS STUDY

⇒ This study includes more than 35 000 patients admitted for strokes of the middle cerebral artery (MCA) between 2016 and 2020.

⇒ Our study attempted to discern the impact of the pandemic state itself from that of concomitant infection with SARS-CoV-2 on the management and outcomes of MCA strokes.

⇒ This study is registry-based and limited by its hospital-based rather than population-based nature.

## INTRODUCTION

Cerebral stroke is a leading cause of morbidity and mortality. In 2019 stroke was globally ranked second as the leading cause of death[1] and disability-adjusted life years in individuals aged above 50.[2] Recent estimates suggest that one in every four individuals is at risk of experiencing a stroke during their lifetime.[3] In addition to the associated mortality and morbidity, stroke is known to generate a significant financial burden on both individual and societal levels.[1 4]

The pandemic caused by the SARS-CoV-2 (COVID-19 pandemic) hitting the world by the end of 2019, was set to impact the healthcare infrastructure in an unprecedented fashion. The COVID-19 pandemic has, directly and indirectly, caused major epidemiological changes to cerebrovascular diseases.[5] While stroke presentations significantly dropped during the pandemic,[6 7] infection with the virus was linked with an increased risk of strokes.[8–11] The pathophysiology

behind this association is thought to be a hypercoagulable state, in turn, linked to abnormal platelet activation, endothelial dysfunction and disruption of the coagulation cascade by the virus.[5 12] Other noteworthy changes during the pandemic were the rise in the prevalence of younger individuals and those with large vessel obstructions among patients with stroke.[13 14] Apart from changes to the epidemiology of stroke, several reports have identified alterations in the use of different treatment modalities during that time.[13 15 16] While many of these changes have been hypothesised to result from altered signalling pathways associated with infection with the virus, altered care-seeking behaviours, and the burden of the pandemic on healthcare infrastructures may also have played a role.[5]

However, few studies on the impact of concomitant SARS-CoV-2 infection on the management and outcomes of stroke have been conducted. Using data provided by the National Inpatient Sample (NIS), the aim of this study was to validate previous reports and investigate the effects of the COVID-19 pandemic and SARS-CoV-2 infection on stroke care and short-term outcomes, focusing on middle cerebral artery (MCA) territory infarctions (figure 1).[17]

## METHODS
### Data source
The NIS, which is maintained by the Healthcare Cost and Utilization Project, is one of the largest inpatient care databases accessible to the public in the USA. The dataset includes approximately 7 million unweighted patient records per year, representing a 20% stratified sample of all Healthcare Cost and Utilization Project community hospitals in the USA. The NIS permits extensive investigations into healthcare utilisation, access, charges, quality, and outcomes and provides dependable national estimates on an annual basis. The data contains a variety of elements, including demographic characteristics, hospital and regional information, diagnoses, procedures, and discharge disposition for all patients for whom documentation exists. More information about the NIS is available online (www.hcup-us.ahrq.gov).

### Cohort selection
Patients diagnosed and treated with ischaemic MCA stroke were identified using the International Classification of Diseases codes 'I66.0', 'I66.01', 'I66.02', 'I66.03', 'I66.09', 'I63.31', 'I633.11', 'I633.13', 'I63.319', 'I63.411', 'I63.412', 'I63.413', 'I63.419', 'I63.51'. 'I63.512', 'I63.513', 'I63.513' and 'I63.519' from the 10th revisions. The study considered three primary treatment modalities: mechanical thrombectomy (MT), intravenous thrombolysis (IVT) or combination of them (MT+IVT). MCA stroke patients who received no treatment were excluded from the study. Initially, 144 486 patients with MCA strokes were identified within the NIS database between 2016 and 2020. However, 64 422 patients were excluded as their treatment approach was not specified, leaving 35 231 patients for inclusion in this study. The study was performed in

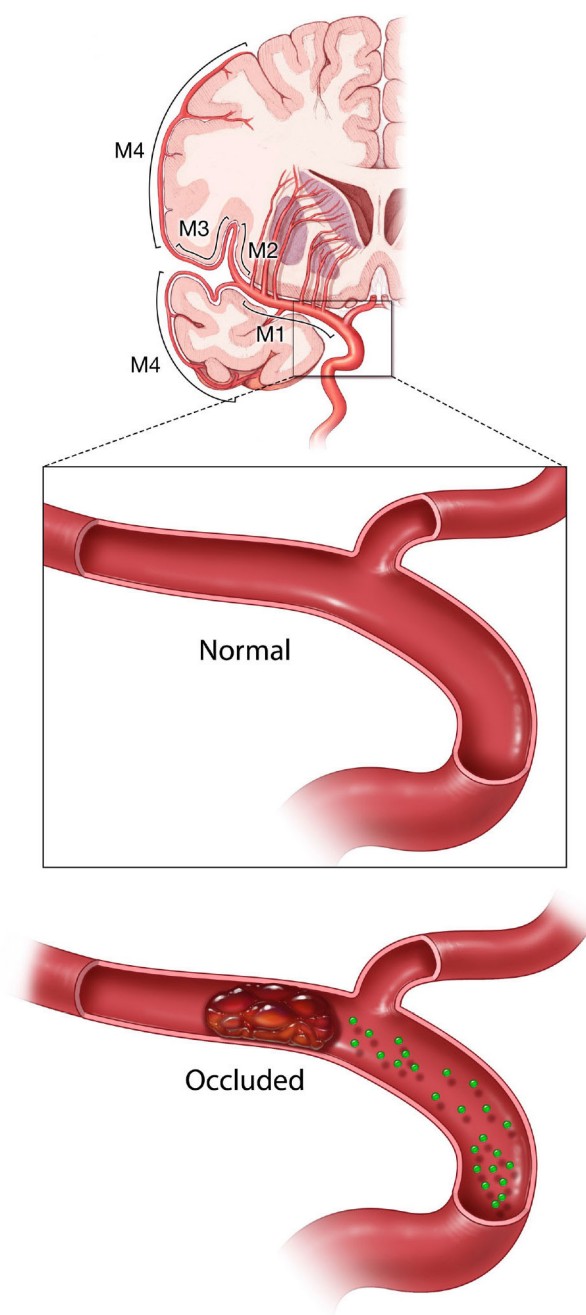

**Figure 1**  Stroke of the middle cerebral artery.

accordance with the Strengthening the Reporting of Observational Studies in Epidemiology guidelines.

### Outcomes and variables
The following variables were extracted: age, gender, race, insurance type, smoking status, history of smoking, hypertension, diabetes and stroke severity measured by National Institute of Health (NIH) Stroke Severity (NIHSS) score. Hospital-related data were recorded, including household income quartile, bed size, location (rural/urban), teaching status, region (Northeast, Midwest, South, West) and control/ownership of the hospital.

The study examined several key outcomes:

► Death, non-routine discharge (transfer to short-term hospital, skilled nursing or intermediate care facility, home healthcare, or against medical advice), length of stay (LOS in days).

► Intervention-related complications such as access-site haemorrhage, subarachnoid haemorrhages (SAHs), vasospasm, intracerebral or intraventricular haemorrhage.

► Medical complications include neurological, cardiac, pulmonary, urinary and thromboembolic events (deep venous thrombosis and pulmonary embolisms).

### Propensity score matching and statistical analysis

Using the Mann-Kendall test, trends regarding the yearly incidence of MCA ischaemic stroke cases, treatment modalities and postprocedural complications between 2016 and 2020 were evaluated.[18] When analysing the impact of concomitant COVID-19 on management and outcomes of MCA strokes, 4:1 propensity score matching based on all available baseline variables, featured in the love plot (online supplemental figure A), was performed, using the K-nearest method with a calliper of 0.2. All statistical analyses were conducted using Python and R software.[19]

### Patient and public involvement

None.

### RESULTS
### Trend analysis

A total of 35 231 patients were included (online supplemental figure B), with 13 465 receiving IVT, 17 060 undergoing MT and 4706 undergoing MT and IVT. Results from the trend analysis revealed that the number of patients with MCA stroke within the NIS database significantly increased between the years 2016 and 2020 (p=0.028).

The proportion of patients receiving IVT decreased significantly (p=0.027) due to a gradual and significant increase in MT (p=0.027). There were no changes in the proportion of patients undergoing both IVT and MT (p=0.086; figure 2). The complications rates remained stable (p>0.05), except for a slight increase in postprocedural SAHs (p=0.043), thromboembolic events (p=0.043) and urinary tract infections (p=0.027; figure 3A,B).

### Impact of the pandemic on management of patients with MCA stroke

There were 8291 patients admitted for MCA stroke the year before the pandemic (2019) and 8812 during the pandemic (2020) (table 1). During 2020, 209 of the patients (2.4%) were concomitantly infected by SARS-CoV-2. There were no differences with respect to demographics, including age, sex, race or ethnicity, primary payer, and hospital location among the two admission periods. During the pandemic, urban, non-teaching hospitals received a larger share of patients with MCA stroke, compared with the year before (9.7% vs 8.5%; p=0.012). The choice of treatment modalities significantly differed between the two admission periods, with MT being more commonly performed during than prior to the pandemic, (p=0.005). The length of hospital stay was similar between the two time periods (mean: 8.2±10.0; p=0.239). Postprocedural complications remained similarly prevalent r, with SAHs occurring in 5.8% of patients, vasospasm in 0.9%, thromboembolic events in 5.3%, neurological complications in 0.5% and cardiac complications in 0.5% of patients (p>0.05). However, access site haemorrhage was more common during the pre-pandemic period (1.2% vs 0.9%; p=0.037), while acute kidney injury (18.5% vs 16.4%; p<0.001), respiratory (6.7% vs 5.9%; p=0.030) and urinary complications (18.5% vs 16.4%; p<0.001) were more common during the pandemic. While the

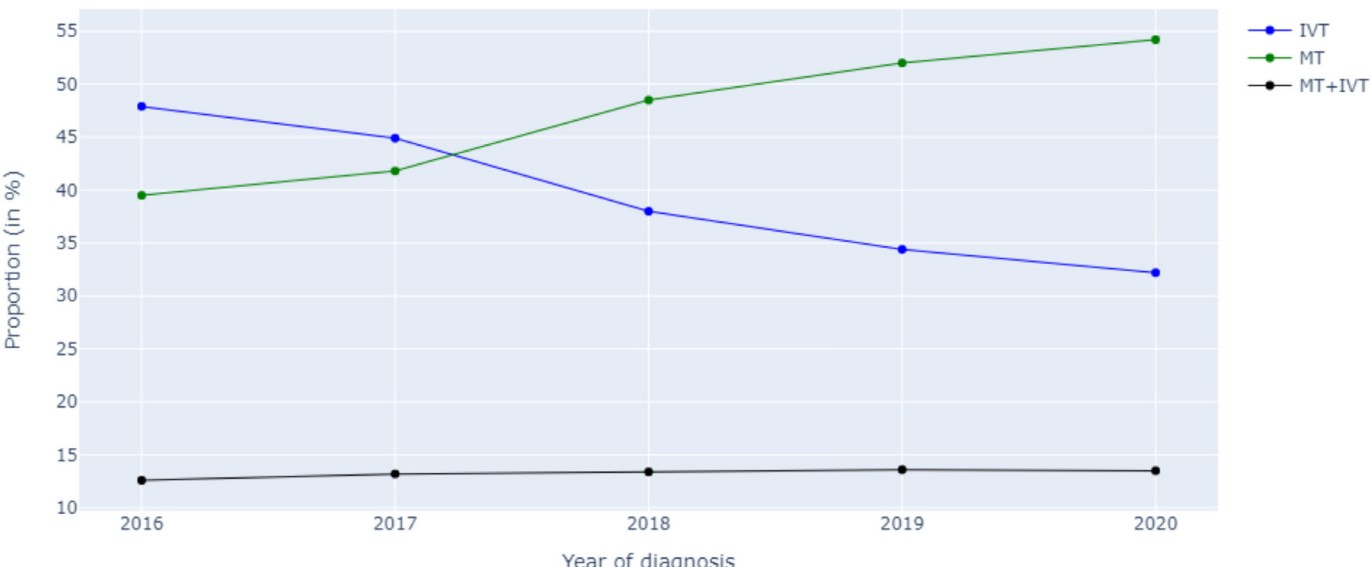

**Figure 2** Trends in the use of the different treatment modalities between 2016 and 2020. IVT, intravenous thrombolysis; MT, mechanical thrombectomy.

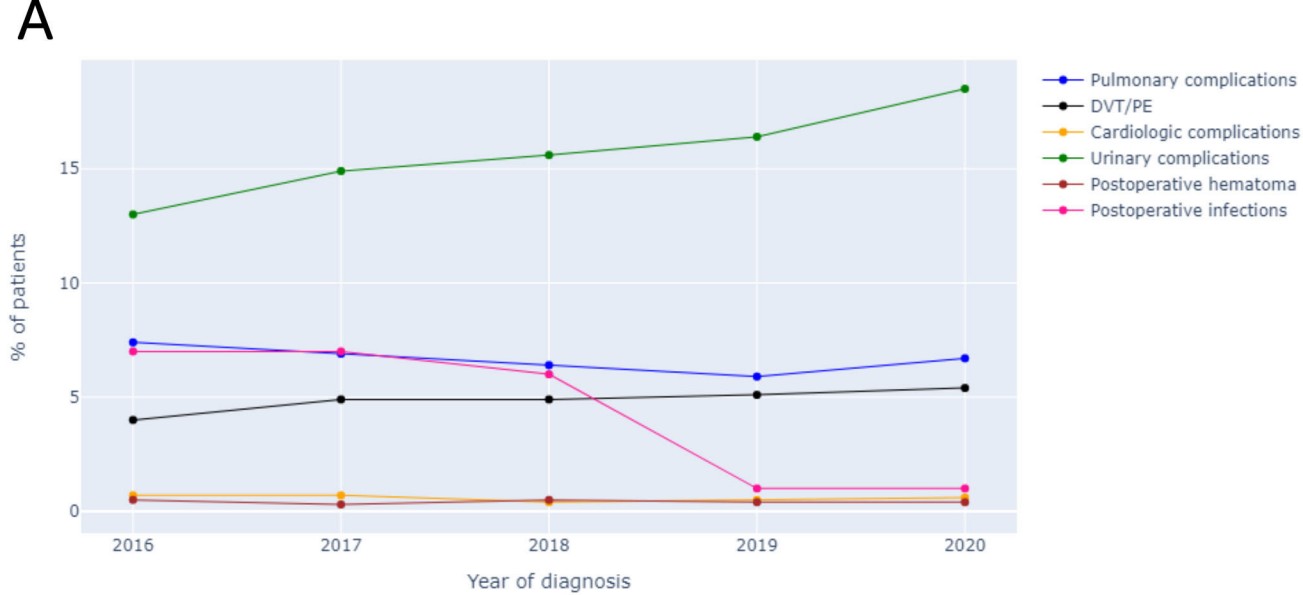

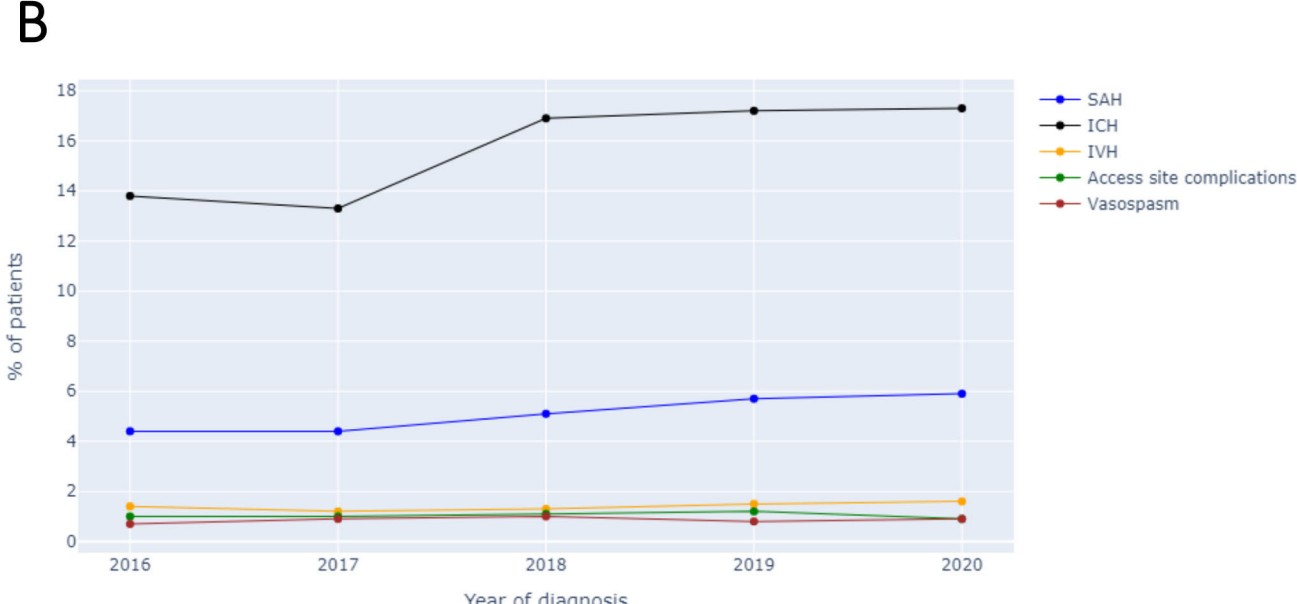

**Figure 3** Trends of (A)postprocedural complications and (B)neurological complications between 2016 and 2020. DVT, deep venous thrombosis; ICH, intracranial haemorrhage; IVH, intraventricular haemorrhage; PE, pulmonary embolisms; SAH, subarachnoid haemorrhage.

proportion of non-routine discharge (76.6%) did not significantly differ between admission periods (p=0.149), there were significantly more in-hospital deaths among patients admitted for MCA stroke during the pandemic (10.2% vs 9.0%; p=0.010) (table 2).

### Impact of concomitant COVID-19 on management and short-term outcomes of MCA stroke

Prior to the propensity score matched analysis, 209 patients with COVID-19 (SARS-CoV-2 positive) were compared with 8603 without COVID-19, admitted during 2020 (table 3). A map showing the distribution of patients with MCA stroke and concomitant COVID-19 is presented (figure 4). Patients with COVID-19 tended to

be males (55% vs 48.8%; p=0.073) and were significantly younger (64.9±14.4 vs 70.0±14.6; p<0.001). White patients with MCA stroke were significantly less likely to present with concomitant COVID-19, opposed to all other races and ethnicities (p<0.001). COVID-19 patients tended to cluster within lower income quartiles (p=0.065) and were significantly more covered by Medicaid (16.35% vs 9.9%; p<0.001). There were no significant differences in hospital region, location, teaching status or number of beds (p>0.05). Patients with COVID-19 had significantly worse NIHSS scores (16–42) (p=0.015). The treatment of choice did not significantly differ between the groups (p=0.562). The mean length of hospital stay

**Table 1** Baseline differences between patients admitted for middle cerebral artery strokes before and during the pandemic (2019 vs 2020)

| | Total (N=17 103) | Pre-pandemic (N=8291) | During the pandemic (N=8812) | P value |
|---|---|---|---|---|
| Female | 8825 (51.6%) | 4322 (52.1%) | 4503 (51.1%) | 0.176 |
| Mean age (SD) | 70.0 (14.6) | 70.2 (14.6) | 69.8 (14.6) | 0.141 |
| Race and ethnicity | | | | 0.124 |
| White | 11 315 (66.2%) | 5545 (66.9%) | 5770 (65.5%) | |
| Black | 2602 (15.2%) | 1232 (14.9%) | 1370 (15.5%) | |
| Hispanic | 1391 (8.1%) | 657 (7.9%) | 734 (8.3%) | |
| Asian or Pacific Islander | 566 (3.3%) | 286 (3.4%) | 280 (3.2%) | |
| Native American | 59 (0.3%) | 32 (0.4%) | 27 (0.3%) | |
| Other | 557 (3.3%) | 269 (3.2%) | 288 (3.3%) | |
| SARS-CoV-2 positive | 209 (1.2%) | n/a | 209 (2.4%) | n/a |
| Income quartile | | | | **0.005** |
| 1 | 4718 (27.6%) | 2295 (27.7%) | 2423 (27.5%) | |
| 2 | 4336 (25.4%) | 2014 (24.3%) | 2322 (26.4%) | |
| 3 | 4209 (24.6%) | 2130 (25.7%) | 2079 (23.6%) | |
| 4 | 3570 (20.9%) | 1722 (20.8%) | 1848 (21.0%) | |
| Primary payer | | | | 0.652 |
| Medicare | 10 822 (63.3%) | 5267 (63.5%) | 5555 (63.0%) | |
| Medicaid | 1659 (9.7%) | 772 (9.3%) | 887 (10.1%) | |
| Private insurance | 3501 (20.5%) | 1715 (20.7%) | 1786 (20.3%) | |
| Self-pay | 648 (3.8%) | 318 (3.8%) | 330 (3.7%) | |
| No charge | 51 (0.3%) | 24 (0.3%) | 27 (0.3%) | |
| Other | 398 (2.3%) | 183 (2.2%) | 215 (2.4%) | |
| Hospital region | | | | 0.439 |
| Northeast | 2975 (17.4%) | 1441 (17.4%) | 1534 (17.4%) | |
| Midwest | 3496 (20.4%) | 1696 (20.5%) | 1800 (20.4%) | |
| South | 6974 (40.8%) | 3421 (41.3%) | 3553 (40.3%) | |
| West | 3658 (21.4%) | 1733 (20.9%) | 1925 (21.8%) | |
| Hospital location and teaching status | | | | **0.012** |
| Rural | 207 (1.2%) | 109 (1.3%) | 98 (1.1%) | |
| Urban non-teaching | 1555 (9.1%) | 702 (8.5%) | 853 (9.7%) | |
| Urban teaching | 15 341 (89.7%) | 7480 (90.2%) | 7861 (89.2%) | |
| Hospital bed size | | | | 0.746 |
| Small | 1560 (9.1%) | 754 (9.1%) | 806 (9.1%) | |
| Medium | 3983 (23.3%) | 1952 (23.5%) | 2031 (23.0%) | |
| Large | 11 560 (67.6%) | 5585 (67.4%) | 5975 (67.8%) | |
| Treatment modality | | | | **0.005** |
| IVT | 5693 (33.3%) | 2855 (34.4%) | 2838 (32.2%) | |
| MT | 9089 (53.1%) | 4309 (52.0%) | 4780 (54.2%) | |
| MT with IVT | 2321 (13.6%) | 1127 (13.6%) | 1194 (13.5%) | |

Bold indicates statistically significant p-values (p<0.05)
IVT, intravenous thrombolysis; MT, mechanical thrombectomy.

was significantly longer among patients with COVID-19 (12.3±12.0 vs 8.2±10.7; p<0.001). The total hospital charges were significantly higher for patients with COVID-19 (p<0.001). Medical, complications, including thromboembolic events, acute kidney failure, respiratory and urinary infection were significantly more prevalent

**Table 2** Outcome differences between patients admitted for middle cerebral artery strokes before and during the pandemic (2019 vs 2020)

| | Total (N=17 103) | Pre-pandemic (N=8291) | During the pandemic (N=8812) | P value |
|---|---|---|---|---|
| Mean length of stay (LOS) in days (SD) | 8.2 (10.0) | 8.1 (9.2) | 8.3 (10.8) | 0.239 |
| Subarachnoid haemorrhage | 993 (5.8%) | 472 (5.7%) | 521 (5.9%) | 0.540 |
| Intracranial haemorrhage | 2945 (17.2%) | 1422 (17.2%) | 1523 (17.3%) | 0.819 |
| Intraventricular haemorrhage | 270 (1.6%) | 128 (1.5%) | 142 (1.6%) | 0.723 |
| Vasospasm | 150 (0.9%) | 70 (0.8%) | 80 (0.9%) | 0.656 |
| Access site haemorrhage | 174 (1.0%) | 98 (1.2%) | 76 (0.9%) | **0.037** |
| Haematoma | 74 (0.4%) | 35 (0.4%) | 39 (0.4%) | 0.839 |
| Wound dehiscence | 15 (0.1%) | 5 (0.1%) | 10 (0.1%) | 0.240 |
| Vascular catheter infection | 12 (0.1%) | 7 (0.1%) | 5 (0.1%) | 0.494 |
| Thromboembolic events | 901 (5.3%) | 424 (5.1%) | 477 (5.4%) | 0.382 |
| Acute kidney injury | 2986 (17.5%) | 1360 (16.4%) | 1626 (18.5%) | **<0.001** |
| Neurological complications | 94 (0.5%) | 48 (0.6%) | 46 (0.5%) | 0.615 |
| Respiratory complications | 1084 (6.3%) | 491 (5.9%) | 593 (6.7%) | **0.030** |
| Cardiac complications | 92 (0.5%) | 43 (0.5%) | 49 (0.6%) | 0.738 |
| Urinary complications | 2986 (17.5%) | 1360 (16.4%) | 1626 (18.5%) | **<0.001** |
| Non-routine discharge | 13 099 (76.6%) | 6310 (76.1%) | 6789 (77.1%) | 0.149 |
| In-hospital mortality | 1646 (9.6%) | 748 (9.0%) | 898 (10.2%) | **0.010** |

Bold indicates statistically significant p-values (p<0.05).

among patients with COVID-19 (p<0.05). Similarly, non-routine discharges (p<0.001) and in-hospital mortality (p<0.001) were significantly more common in this patient group.

After propensity score matching using patient demographics, baseline characteristics and comorbidities, and stroke severity on the NIHSS scale (online supplemental figure A), only the rate of thromboembolic events (p=0.035), respiratory complications (p=0.029) and in-hospital mortality (p<0.001) remained significant. This suggests a direct correlation between COVID-19 and the occurrence of complications and as in-hospital mortality. This was further verified using multivariable logistic regression, which indicated that concomitant infection with SARS-CoV-2 was a significant and independent predictor of in-hospital mortality (OR: 3.5; 95% CI 2.9–4.0; p<0.01). Surprisingly, COVID-19 patients were seemingly less affected by haemorrhages, in particular intracranial ones, compared with patients without COVID-19 (23.5% vs 15.4%; p=0.014) (table 3).

## DISCUSSION
In this study, 35 231 patients with MCA stroke between 2016 and 2020 were included. Results from the trend analysis revealed that the number of patients with MCA stroke within the NIS database gradually increased during the study period including the pandemic (2020). This is in opposition to previous reports showing a decrease in stroke admissions during the pandemic. Global reports have indicated a decline in the volume of stroke

hospitalisations, with primary stroke centres and centres with higher COVID-19 inpatient volumes experiencing steeper declines.[5 6 15] A recovery of initial stroke hospitalisation rates was not witnessed until later during the pandemic.[15] The results of this study mainly reflect the coverage of the NIS registry and may merely indicate an expansion of the database during that time, rather than an actual increase in stroke admissions. The proportion of patients receiving IVT significantly decreased during the study period (48%–32%; p=0.027), due to a gradual and significant increase in MT (40%–55%; p=0.027).

To our knowledge, there are no previous works contrasting the impact of concomitant infection with SARS-CoV-2 with the impact of the unique circumstances created by the pandemic itself on the management of MCA strokes.

In this study, patients with COVID-19 were younger (p<0.001) but had significantly worse NIHSS scores (16-42). A meta-analysis of 129 491 patients reached similar conclusions, revealing that patients who were admitted for stroke, were significantly younger and had strokes of higher severity grades. There have been several reports highlighting the occurrence of large-vessel occlusions in young patients with COVID-19.[14 20] In one study, patients with stroke and concomitant COVID-19 were significantly younger and lacked vascular risk factors. Despite being healthier at baseline, these patients had poorer outcomes and were less likely to experience complete revascularisation compared with matched controls without COVID-19.[14] The occurrence of larger strokes in this group of

**Table 3** Differences between COVID-19 positive and negative patients admitted for middle cerebral artery strokes during the pandemic year of 2020

| | Pre-matching analysis | | | Propensity score matched analysis | | |
|---|---|---|---|---|---|---|
| | SARS-CoV-2 positive (N=209) | SARS-CoV-2 negative (N=8603) | P value | SARS-CoV-2 positive (N=195) | SARS-CoV-2 negative (N=753) | P value |
| Female sex | 94 (45.0%) | 4409 (51.2%) | 0.073 | 90 (46.2%) | 346 (45.9%) | 0.959 |
| Mean age (SD) | 64.9 (14.4) | 70.0 (14.6) | **<0.001** | 65.4 (14.3) | 65.7 (15.7) | 0.773 |
| NIHSS score | | | **0.015** | | | 0.991 |
| 1–4 | 8 (3.8%) | 918 (10.7%) | | 8 (4.1%) | 31 (4.1%) | |
| 5–15 | 66 (31.6%) | 3121 (36.3%) | | 65 (33.3%) | 249 (33.1%) | |
| 16–20 | 34 (16.3%) | 1302 (15.1%) | | 31 (15.9%) | 125 (16.6%) | |
| 21–42 | 42 (20.1%) | 1534 (17.8%) | | 36 (18.5%) | 148 (19.7%) | |
| N-miss | 59 (28.2%) | 1728 (20.1%) | | 55 (28.2%) | 200 (26.6%) | |
| Treatment modality | | | 0.562 | | | 0.979 |
| IVT | 65 (31.1%) | 2773 (32.2%) | | 62 (31.8%) | 245 (32.5%) | |
| MT | 120 (57.4%) | 4660 (54.2%) | | 110 (56.4%) | 419 (55.6%) | |
| MT with IVT | 24 (11.5%) | 1170 (13.6%) | | 23 (11.8%) | 89 (11.8%) | |
| Outcomes | | | | | | |
| Mean length of stay (LOS) in days (SD) | 12.3 (12.0) | 8.2 (10.7) | **<0.001** | 10.6 (8.6) | 10.4 (9.0) | 0.788 |
| Mean total charges in USD (SD) | 255 920 (238 810) | 192 269 (183 666) | **<0.001** | 229 414 (193 518) | 231 857 (196 519) | 0.878 |
| Subarachnoid haemorrhage | 16 (7.7%) | 505 (5.9%) | 0.280 | 14 (7.2%) | 53 (7.0%) | 0.945 |
| Intracranial haemorrhage | 33 (15.8%) | 1490 (17.3%) | 0.563 | 30 (15.4%) | 177 (23.5%) | **0.014** |
| Intraventricular haemorrhage | 2 (1.0%) | 140 (1.6%) | 0.447 | 2 (1.0%) | 14 (1.9%) | 0.421 |
| Access site haemorrhage | 0 (0.0%) | 76 (0.9%) | 0.172 | 0 (0.0%) | 3 (0.4%) | 0.377 |
| Vasospasm | 2 (1.0%) | 78 (0.9%) | 0.940 | 2 (1.0%) | 11 (1.5%) | 0.641 |
| Haemorrhagic stroke | 18 (8.6%) | 710 (8.3%) | 0.852 | 18 (9.2%) | 83 (11.0%) | 0.470 |
| Wound dehiscence | 1 (0.5%) | 9 (0.1%) | 0.113 | 1 (0.5%) | 2 (0.3%) | 0.584 |
| Vascular catheter infection | 0 (0.0%) | 5 (0.1%) | 0.727 | 0 (0.0%) | 2 (0.3%) | 0.471 |
| Thromboembolic events | 25 (12.0%) | 452 (5.3%) | **<0.001** | 24 (12.3%) | 57 (7.6%) | **0.035** |
| Acute kidney injury | 56 (26.8%) | 1570 (18.2%) | **0.002** | 49 (25.1%) | 152 (20.2%) | 0.132 |
| Myocardial infarction | 15 (7.2%) | 487 (5.7%) | 0.350 | 12 (6.2%) | 47 (6.2%) | 0.964 |
| Neurological complications | 0 (0.0%) | 46 (0.5%) | 0.289 | 0 (0.0%) | 7 (0.9%) | 0.177 |
| Respiratory complications | 26 (12.4%) | 567 (6.6%) | **<0.001** | 22 (11.3%) | 50 (6.6%) | **0.029** |
| Cardiac complications | 0 (0.0%) | 49 (0.6%) | 0.274 | 0 (0.0%) | 5 (0.7%) | 0.254 |
| Urinary complications | 56 (26.8%) | 1570 (18.2%) | **0.002** | 49 (25.1%) | 152 (20.2%) | 0.132 |
| Non-routine discharge | 176 (84.2%) | 6613 (76.9%) | **0.013** | 165 (84.6%) | 615 (81.8%) | 0.355 |
| In-hospital mortality | 55 (26.3%) | 843 (9.8%) | **<0.001** | 52 (26.7%) | 64 (8.5%) | **<0.001** |

Bold indicates statistically significant p-values (p<0.05).
IVT, intravenous thrombolysis; MT, mechanical thrombectomy; NIHSS, NIH Stroke Severity.

younger and healthier patients was hypothesised to be due to the hypercoagulable state associated with infection with SARS-CoV-2, leading to thrombosis.[5 12 21 22]

Comparing the pre-pandemic year of 2019 with the year of 2020 during which the first wave of the pandemic occurred, we found that MT was more commonly

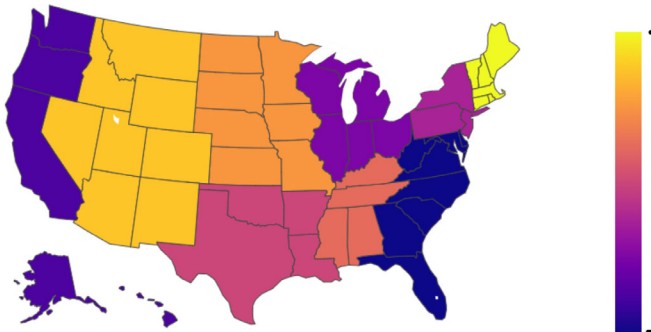

**Figure 4** Map of the USA showing the distribution of patients with middle cerebral artery strokes and concomitant COVID-19.

performed than IVT (p=0.005). Although this discrepancy may be explained by the fact that large vessel occlusions occurred more frequently among patients with COVID-19,[5 13 23] our results did not reveal any significant difference in treatment modality between patients with and without COVID-19. It is important to note, though, that we cannot completely disregard the possibility that the observed difference might be a continuation of a pre-existing trend towards a gradual increase in the use of MT over the years.

Also, a significantly increased risk of thromboembolic events in patients with MCA strokes and concomitant COVID-19, was found in the current study. This highlights the well-established correlation between COVID-19 and the occurrence of thromboembolic events,[24 25] which lead to recommendations for prophylactic treatment of hospitalised COVID-19 patients with anticoagulation therapy.[26] Surprisingly, post-matching results indicated that 23.5% of patients without COVID-19 experienced intracranial haemorrhage compared with 15.4% of those with COVID-19 (p=0.014). We hypothesise that this effect may have been the result of the hypercoagulable state associated with COVID-19.[5 12 21 22] Although previous reports seem to suggest the opposite; with increased rates of intracranial haemorrhage among patients with COVID-19, this association was mainly related to the treatment with anticoagulation therapy in this group of patients.[27] Additionally, the study period only considers the first year of the pandemic, during which recommendations regarding prophylactic anticoagulation therapy still were not generalised. This may explain the lower haemorrhage risk among these patients.[15]

Finally, our study revealed a slightly higher in-hospital mortality rate during the first year of the pandemic as compared with the year before that (p=0.010). This may be due to a number of factors. First, the strain on the healthcare systems at that time, may have created disruptions in routine care leading to suboptimal treatment of this patient group. Additionally, the fear of contracting COVID-19 as well as the public health measures undertaken during this period may have led to delayed medical attention for patients who had a stroke. This may have contributed to more severe cases of stroke and in turn a

higher mortality. Also, concomitant infection with SARS-CoV-2 itself may have contributed to the increased death toll through larger strokes, as previously mentioned, or an increased rate of adverse events. In fact, longer LOSs (p<0.001), risk of non-routine discharge (p=0.013) and higher in-hospital mortality rate (p<0.001) were all noted among patients with MCA stroke and concomitant COVID-19, which may have been the result of higher stroke severity (NIHSS scores) in this group. After adjusting for confounders, including the NIHSS scores through propensity score matching, only in-hospital mortality remained significant (p<0.001). This direct correlation between COVID-19 and in-hospital mortality was likely the result of the increased prevalence of thromboembolic events (p=0.035) and respiratory complications (p=0.029) witnessed in patients with COVID-19.

Nonetheless, this study has several limitations. Primarily, the sample size of patients with concomitant COVID-19 infection was very small compared with the whole cohort of patients with MCA stroke. The data provided by the NIS is limited by its hospital-based rather than population-based nature. Additionally, as with all registry-based studies, there is a risk of reporting and coding biases, loss to follow-up, and attrition, as well as other weaknesses. The NIS also lacks clinically important endpoints, such as patient-reported, health-related quality of life, neurological and long-term clinical outcomes, as well as granularity in terms of the cause of death and other epidemiological elements. Moreover, the registry only targets the US population which limits international generalisability and calls for external validation of the findings. Although propensity score matching was performed using all baseline data available on hand, the absence of other potential confounding variables that are not captured by the NIS; including various comorbidities, limited our ability to pinpoint the true effect of COVID-19 on the outcomes of interest.

## CONCLUSION

Among patients with MCA stroke, those with concomitant COVID-19 were significantly younger and had higher stroke severity scores on the NIHSS scale. They were more likely to experience thromboembolic complications and in-hospital mortality compared with matched controls.

**Author affiliations**
[1]Mayo Clinic, Rochester, Minnesota, USA
[2]Clinical Neuroscience, Karolinska Institute, Stockholm, Sweden
[3]Neurological Surgery, Thomas Jefferson University Hospital, Philadelphia, Pennsylvania, USA
[4]Mayo Clinic in Florida, Jacksonville, Florida, USA
[5]Neurosurgery, Thomas Jefferson University Hospital, Philadelphia, Pennsylvania, USA

**Contributors** AKG, VGE-H: conception and design of the work, drafting of the article, critical revision, and final approval of the version to be published. EA, JRZ, KR, MG, HH: conception and design of the work, drafting of the article, and final approval of the version to be published. MB, MO, AE-T, TR: conception and design of the work, drafting of the article, critical revision, and final approval of the version

to be published. PJ: guarantor of the review, conception and design of the work, drafting of the article, critical revision, and final approval of the version to be published.

**Funding** The authors have not declared a specific grant for this research from any funding agency in the public, commercial or not-for-profit sectors.

**Map disclaimer** The depiction of boundaries on the map(s) in this article does not imply the expression of any opinion whatsoever on the part of BMJ (or any member of its group) concerning the legal status of any country, territory, jurisdiction or area or of its authorities. The map(s) are provided without any warranty of any kind, either express or implied.

**Competing interests** None declared.

**Patient and public involvement** Patients and/or the public were not involved in the design, or conduct, or reporting, or dissemination plans of this research.

**Patient consent for publication** Not applicable.

**Ethics approval** This study was performed in accordance with all ethical guidelines. The need for ethical approval was waived by the Mayo IRB due to the deidentified nature of the NIS database. Consent to participate is not applicable.

**Provenance and peer review** Not commissioned; externally peer reviewed.

**Data availability statement** Data are available upon reasonable request. Data may be provided upon reasonable request. Our data originates from the NIS database.

**ORCID iDs**
Victor Gabriel El-Hajj http://orcid.org/0000-0001-9479-761X
Jorge Rios Zermeno http://orcid.org/0000-0002-5239-7041
Adrian Elmi-Terander http://orcid.org/0000-0002-3776-6136

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
