## [Reviewer comments · BMJ Open]

This paper was submitted to a another journal from BMJ but declined for publication following peer review. The authors addressed the reviewers' comments and submitted the revised paper to BMJ Open. The paper was subsequently accepted for publication at BMJ Open.

ARTICLE DETAILS

TITLE (PROVISIONAL)	Impact of the pandemic and concomitant COVID-19 on the management and outcomes of middle cerebral artery strokes: a nationwide registry-based study
AUTHORS	Ghaith, Abdul Karim; El-Hajj, Victor Gabriel; Atallah, Elias; Rios Zermeno, Jorge; Ravindran, Krishnan; Gharios, Maria; Hoang, Harry; Bydon, Mohamad; Ohlsson, Marcus; Elmi-Terander, Adrian; Rabih, Tawk; Jabbour, Pascal

VERSION 1 – REVIEW

REVIEWER	Huo, Xiaochuan Beijing Tiantan Hospital, Interventional Neurology
REVIEW RETURNED	19-Nov-2023

GENERAL COMMENTS	This study presents a compelling exploration of the impact of the pandemic and concurrent COVID-19 on the management and outcomes of Middle Cerebral Artery (MCA) strokes in the United States. It encompasses a substantial cohort of 35,231 MCA stroke patients, including 209 individuals who were concurrently diagnosed with COVID-19. The study's comparison of characteristics between patients with and without COVID-19 underpins the significant finding that those with concurrent COVID-19 were notably younger and exhibited higher stroke severity, as measured by the NIHSS scale. Additionally, they were more susceptible to thromboembolic complications and in-hospital mortality in comparison to their matched counterparts. Despite these insights, I believe the study falls short of meeting the standard criteria for publication. The following points warrant the authors' careful consideration. Comments: 1, This study lacks a transparent patient selection process, despite mentioning that patients with Middle Cerebral Artery (MCA) strokes were identified within the NIS database from 2016 to 2020. The basis for determining the specific time interval for the study remains unclear. Furthermore, well-defined inclusion and exclusion criteria are crucial, particularly as comorbid conditions, such as other malignant diseases, could significantly influence patient outcomes. It is essential for the authors to clarify these aspects to enhance the study's robustness and reliability.2, Regarding the selection of treatment modalities, the study presents a comparison between the pre-pandemic year of 2019
--

	and the year 2020, during the initial wave of the pandemic. It was observed that Mechanical Thrombectomy (MT) was more frequently administered compared to Intravenous Thrombolysis (IVT), with a significant difference noted ($p = 0.005$). However, the study did not find a notable difference in the choice of treatment modality between patients with and without COVID-19. This raises a crucial question: Did COVID-19 influence the decision-making process regarding treatment modalities? Clarification on this aspect is needed to understand the impact of COVID-19 on treatment choices for MCA strokes. 3, In this study, a clear elucidation of the criteria for selecting treatment modalities (Mechanical Thrombectomy [MT], Intravenous Thrombolysis [IVT], or a combination of both [MT+IVT]) for patients with Middle Cerebral Artery (MCA) strokes is essential to mitigate any potential bias in patient selection. It is also important to note that associated studies have supported the premise that pre-IVT administration offers additional benefits to MT in terms of clinical and imaging outcomes for acute ischemic strokes caused by large vessel occlusions, without a concomitant increase in symptomatic intracerebral hemorrhage. Could the differences in treatment modalities between patients with and without COVID-19 have contributed to the incidence of procedure-related complications? 4, In this study, it is imperative to establish a clear methodology for distinguishing patients who developed Middle Cerebral Artery (MCA) strokes as a secondary complication of COVID-19 from those who contracted COVID-19 following an MCA stroke. The divergence in the disease progression pathways could be closely tied to the clinical outcomes of the patients enrolled. Therefore, a thorough understanding of the patients' medical history prior to undergoing treatment holds significant relevance in evaluating and interpreting the study's findings. 5, In this study, it was observed that among patients with Middle Cerebral Artery (MCA) strokes, those concurrently diagnosed with COVID-19 exhibited a higher likelihood of experiencing thromboembolic complications and in-hospital mortality when compared to their matched counterparts. The observed increase in in-hospital mortality among these patients may predominantly be associated with COVID-19. Consequently, it becomes essential to delineate and explain the differences in the clinical profiles and outcomes of patients admitted for MCA strokes prior to and during the pandemic. Such an analysis is crucial for a comprehensive understanding of the impact of COVID-19 on the severity and management of MCA strokes. 6, There were spelling errors that need to be revised in the text, such as 'NHSS scores'
--	--

REVIEWER	Wang, Chuanying
REVIEW RETURNED	02-Dec-2023

GENERAL COMMENTS	After revising, this paper is better than before. However, it still has several issues should be addressed.  - The abstract section In the conclusion part, the paper writes “They were more likely to experience thromboembolic complications and in-hospital mortality compared to matched controls”, but readers can't find any information about thromboembolic in the method and result part.  - The introduction section
--

	The description of stroke epidemiology in the first paragraph is still redundant and needs to be appropriately deleted.  - The method section The authors should give the definition of “respiratory complication” and “urinary complication”. - The result section It is recommended that the authors add odds ratio and 95% confidence interval of multivariable logistic regression. In addition, the font of this sentence should be consistent with others. - The discussion section The authors explained the direct correlation between COVID-19 and in-hospital mortality was likely the result of the increased prevalence of thromboembolic events and respiratory complications. I would suggest the authors add mediation analysis to testify this possible causality. - Please carefully check word spelling in text, such as Page 38 line 36 “stoke” should be “stroke”.
--	---

VERSION 1 – AUTHOR RESPONSE

Reviewer: 1

Dr. Xiaochuan Huo, Beijing Tiantan Hospital

Comments to the Author:

This study presents a compelling exploration of the impact of the pandemic and concurrent COVID-19 on the management and outcomes of Middle Cerebral Artery (MCA) strokes in the United States. It encompasses a substantial cohort of 35,231 MCA stroke patients, including 209 individuals who were concurrently diagnosed with COVID-19. The study's comparison of characteristics between patients with and without COVID-19 underpins the significant finding that those with concurrent COVID-19 were notably younger and exhibited higher stroke severity, as measured by the NIHSS scale. Additionally, they were more susceptible to thromboembolic complications and in-hospital mortality in comparison to their matched counterparts. Despite these insights, I believe the study falls short of meeting the standard criteria for publication. The following points warrant the authors' careful consideration.

Comments:

1, This study lacks a transparent patient selection process, despite mentioning that patients with Middle Cerebral Artery (MCA) strokes were identified within the NIS database from 2016 to 2020. The basis for determining the specific time interval for the study remains unclear. Furthermore, well-defined inclusion and exclusion criteria are crucial, particularly as comorbid conditions, such as other malignant diseases, could significantly influence patient outcomes. It is essential for the authors to clarify these aspects to enhance the study's robustness and reliability.

Thank you for this question. Access to such databases costs money. With the funds we have we could only get access to the 2016-2020 period. Additionally, more recent years were not available at the time of the writing of this manuscript. There were no exclusion criteria. Only patients lacking outcome data (missing data) for complications or any of the study outcomes were naturally excluded. Otherwise, all patients were included to avoid selection biases.

2, Regarding the selection of treatment modalities, the study presents a comparison between the pre-pandemic year of 2019 and the year 2020, during the initial wave of the pandemic. It was observed that Mechanical Thrombectomy (MT) was more frequently administered compared to Intravenous Thrombolysis (IVT), with a significant difference noted ($p = 0.005$). However, the study did not find a notable difference in the choice of treatment modality between patients with and without COVID-19. This raises a crucial question: Did COVID-19 influence the decision-making process regarding treatment modalities? Clarification on this aspect is needed to understand the impact of COVID-19 on treatment choices for MCA strokes.

Thank you for bringing this observation to our attention. According to our data, the choice of treatment does not appear to be directly influenced by COVID-19 itself. However, there is an observed shift towards MT during the pandemic. Our hypothesis is hence that the specific challenges imposed on healthcare infrastructures by the pandemic may contribute to this trend. It is important to note, though, that we cannot completely disregard the possibility that the observed difference might be a continuation of a pre-existing trend favoring the gradual increase in the use of MT as opposed to IVT over the years. We have now added a statement to the discussion to highlight that: It is important to note, though, that we cannot completely disregard the possibility that the observed difference might be a continuation of a pre-existing trend favoring the gradual increase in the use of MT as opposed to IVT over the years.

3, In this study, a clear elucidation of the criteria for selecting treatment modalities (Mechanical Thrombectomy [MT], Intravenous Thrombolysis [IVT], or a combination of both [MT+IVT]) for patients with Middle Cerebral Artery (MCA) strokes is essential to mitigate any potential bias in patient selection. It is also important to note that associated studies have supported the premise that pre-IVT administration offers additional benefits to MT in terms of clinical and imaging outcomes for acute ischemic strokes caused by large vessel occlusions, without a concomitant increase in symptomatic intracerebral hemorrhage. Could the differences in treatment modalities between patients with and without COVID-19 have contributed to the incidence of procedure-related complications?

Thank you for raising this question. We consider it highly improbable that the variance in procedure-related complications was influenced by differences in treatment modalities. Primarily, this is because there were no significant distinctions in treatment modalities between Covid-positive and negative patients ($p=0.562$). While we acknowledge the presence of some minor differences (given that the p-value is not approximately 1), it's important to note that propensity score matching was conducted to address these variations. Subsequent to the matching procedure, there were virtually no differences between the groups ($p=0.98$).

4, In this study, it is imperative to establish a clear methodology for distinguishing patients who developed Middle Cerebral Artery (MCA) strokes as a secondary complication of COVID-19 from those who contracted COVID-19 following an MCA stroke. The divergence in the disease progression pathways could be closely tied to the clinical outcomes of the patients enrolled. Therefore, a thorough understanding of the patients' medical history prior to undergoing treatment holds significant relevance in evaluating and interpreting the study's findings.

While we acknowledge the relevance of what the reviewer states, we unfortunately do not hold precise charting data, since the present study is a database review from the NIS. We can unfortunately only work with what the database offers. In corroboration with the reviewer's view, we have added this aspect to the limitation section:

The NIS also lacks clinically important endpoints, such as patient-reported, health-related quality of life, neurological, and long-term clinical outcomes, as well as granularity in terms of the cause of death and other epidemiological elements.

5, In this study, it was observed that among patients with Middle Cerebral Artery (MCA) strokes, those concurrently diagnosed with COVID-19 exhibited a higher likelihood of experiencing thromboembolic complications and in-hospital mortality when compared to their matched counterparts. The observed increase in in-hospital mortality among these patients may predominantly be associated with COVID-19. Consequently, it becomes essential to delineate and explain the differences in the clinical profiles and outcomes of patients admitted for MCA strokes prior to and during the pandemic. Such an analysis is crucial for a comprehensive understanding of the impact of COVID-19 on the severity and management of MCA strokes.

We express our gratitude to the reviewer for their valuable input. All baseline variables from the NIS database have been incorporated into the analysis and are comprehensively presented in tables 1 and 2. Moreover, propensity scores derived from these baseline variables were utilized to effectively mitigate any confounding factors or external influences that could introduce bias into the analysis. This comprehensive matching process even considered stroke severity (NIHSS) scores, age, and other factors that could ultimately influence the in-hospital complication and mortality risks. Both pre- and post-matching analyses are presented in order to offer a nuanced view into the impact of both COVID-19 itself and other baseline variables on the outcomes of interest.

With a relatively high degree of confidence, given the data, we can hence say that the observed increase in in-hospital mortality and complications among patients with concomitant COVID-19 is likely attributable to the virus itself, rather than any of the other confounding variables. However, it is essential to acknowledge that this statement is subject to certain limitations, as the NIS database may not capture some unknown confounders that could still be present. This was added to the limitation section:

Although propensity score matching was performed using all baseline data available on hand, the absence of other potential confounding variables that are not captured by the NIS; including various comorbidities, limited our ability to pinpoint the true effect of COVID-19 on the outcomes of interest.

6, There were spelling errors that need to be revised in the text, such as 'NHSS scores'.

We thank the reviewer for this astute observation. We have now controlled the text and revised the spelling mistakes.

Reviewer: 2

Dr. Chuanying Wang

Comments to the Author:

After revising, this paper is better than before. However, it still has several issues should be addressed.

- The abstract section

In the conclusion part, the paper writes "They were more likely to experience thromboembolic complications and in-hospital mortality compared to matched controls", but readers can't find any information about thromboembolic in the method and result part.

Thank you for this observation. We have now added the missing text as requested:

Additionally, both pre- and post-matching patients with COVID-19 were more likely to experience thromboembolic ($p=0.035$), and respiratory complications ($p=0.029$).

- The introduction section

The description of stroke epidemiology in the first paragraph is still redundant and needs to be appropriately deleted.

Thank you for this remark. We have now significantly shortened the introduction accordingly.

- The method section

The authors should give the definition of “respiratory complication” and “urinary complication”.

Thank you for this observation. These variables are present in the NIS database and mainly represent lower respiratory tract infections, pneumonia, respiratory distress syndrome, respiratory failure, intubation-related issues, airway damage, and urinary tract infections, Foley-catheter related issues, postoperative hematuria, respectively. These explanations have now been added as follows: Respiratory complications included: lower respiratory tract infections, respiratory distress syndrome, respiratory failure, intubation-related complications, airway damage, while urinary complications included: urinary tract infections, Foley-catheter-related issues, and postoperative hematuria.

- The result section

It is recommended that the authors add odds ratio and 95% confidence interval of multivariable logistic regression. In addition, the font of this sentence should be consistent with others.

Thank you for this astute observation. The requested changes have made:

(OR: 3.5; CI95%: 2.9-4.0; $p < 0.01$).

- The discussion section

The authors explained the direct correlation between COVID-19 and in-hospital mortality was likely the result of the increased prevalence of thromboembolic events and respiratory complications. I would suggest the authors add mediation analysis to testify this possible causality.

Thank you for this suggestion. We are unfortunately not familiar with this technique. Since, we our team lacks a statistician, our ability to perform complex statistical analyses is unfortunately limited. We hope the reviewer understands. Nonetheless, our statement regarding this plausible causation is still valid, based on established evidence showing that thromboembolic events as well as respiratory complications are major contributors of postoperative mortality. Hence, since the rate of these complications is significantly higher among patients with COVID-19, it is not unreasonable to assume a causal relationship between these complications and mortality among these patients.

- Please carefully check word spelling in text, such as Page 38 line 36 “stoke” should be “stroke”.

Again, thank you this astute observation. This has now been modified.

Editor(s)' Comments to the Authors:

*Please revise the title of your manuscript to include the research question, study design and setting. This is the preferred format of the journal. Ideally this would take the form ‘Study aim: study design and setting’ (please see published examples). Please also ensure that the article title uses sentence case instead of capitalising every word.

Thank you. This was done.

*Please revise the abstract to ensure that it is formatted according to our Instructions for Authors (<http://bmjopen.bmj.com/pages/authors/#research>), including all relevant subheadings and required details.

We have now modified that accordingly.

*Please update the abstract Results section to include full numerical data for the findings reported (it is not generally helpful to report only p values – where possible, the magnitude data to which these p values apply should also be reported to allow interpretation.
The numbers were now added.

*Please change the heading ‘Strengths and limitations’ to ‘Strengths and limitations of this study’.
This was now done.

*Please move the ‘Patient and public involvement’ statement to the end of the Methods (immediately before the Results section) and simplify the statement to read “None.”
Thank You for your assistance. Done.

*Please remove the ‘Limitations’ subheading within the Discussion – the text on limitations should be included within the Discussion, rather than under a separate heading.
This was done as requested.

*Please move the ‘Data availability statement’ (currently at the start of the manuscript) to the end of the manuscript, alongside the other statements (immediately before the reference list).

Thank you for this remark. Done.

*Please change the heading ‘Author Contributions’ to ‘Contributors’ and please delete the ‘Acknowledgements’ statement (since it is not required).

Done.

*In the ‘Ethics approval and consent to participate’ statement, please correct ‘waved’ to waived’.

Done. Thank you.

VERSION 2 – REVIEW

REVIEWER	Huo, Xiaochuan Beijing Tiantan Hospital, Interventional Neurology
REVIEW RETURNED	15-Feb-2024

GENERAL COMMENTS	The paper “impact of the pandemic and concomitant COVID-19 on the management and outcomes of middle cerebral artery strokes: a nationwide registry-based study” is well written and well revised.
---

REVIEWER	Wang, Chuanying
REVIEW RETURNED	15-Jan-2024

GENERAL COMMENTS	None.
-------